# The Promising Nanovectors for Gene Delivery in Plant Genome Engineering

**DOI:** 10.3390/ijms23158501

**Published:** 2022-07-31

**Authors:** Heng Zhi, Shengen Zhou, Wenbo Pan, Yun Shang, Zhanghua Zeng, Huawei Zhang

**Affiliations:** 1School of Advanced Agricultural Sciences, Peking University, Beijing 100871, China; heng.zhi@pku-iaas.edu.cn (H.Z.); shengen.zhou@pku-iaas.edu.cn (S.Z.); wenbo.pan@pku-iaas.edu.cn (W.P.); 2Institute of Advanced Agricultural Science, Peking University, Weifang 261000, China; yun.shang@pku-iaas.edu.cn; 3Institute of Environment and Sustainable Development in Agriculture, CAAS Chinese Academy of Agricultural Science, Beijing 100081, China; zengzhanghua@caas.cn

**Keywords:** gene delivery, transformation, genetic engineering, nanomaterials, crop breeding

## Abstract

Highly efficient gene delivery systems are essential for genetic engineering in plants. Traditional delivery methods have been widely used, such as Agrobacterium-mediated transformation, polyethylene glycol (PEG)-mediated delivery, biolistic particle bombardment, and viral transfection. However, genotype dependence and other drawbacks of these techniques limit the application of genetic engineering, particularly genome editing in many crop plants. There is a great need to develop newer gene delivery vectors or methods. Recently, nanomaterials such as mesoporous silica particles (MSNs), AuNPs, carbon nanotubes (CNTs), and layer double hydroxides (LDHs), have emerged as promising vectors for the delivery of genome engineering tools (DNA, RNA, proteins, and RNPs) to plants in a species-independent manner with high efficiency. Some exciting results have been reported, such as the successful delivery of cargo genes into plants and the generation of genome stable transgenic cotton and maize plants, which have provided some new routines for genome engineering in plants. Thus, in this review, we summarized recent progress in the utilization of nanomaterials for plant genetic transformation and discussed the advantages and limitations of different methods. Furthermore, we emphasized the advantages and potential broad applications of nanomaterials in plant genome editing, which provides guidance for future applications of nanomaterials in plant genetic engineering and crop breeding.

## 1. Introduction

Climate change and rapidly increasing human populations pose challenges to ensuring food security worldwide [1]. Plant genetic engineering, which is also known as genetic transformation or genomic manipulation, is a key strategy for breeding crops with valuable traits such as increased yield and nutrition value, enhanced tolerance to biotic and abiotic stresses, efficient nutrient uptake, and herbicide tolerance to meet future demands [2]. Delivering genes of interest from other species into a plant genome by transient expression or stable integration enables attainment of desired agronomic traits [3].

Highly efficient gene delivery methods are essential for genetic engineering. Traditional delivery systems include Agrobacterium-mediated transformation, biolistic particle bombardment, polyethylene glycol (PEG)-mediated transfection, and viral transfection. These systems have been established in many plant species, including staple crops, fruits, vegetables, and even trees. However, strong genotype dependence and highly technical requirements of these systems limit their broad application [4,5]. In recent years, the utilization of developmental regulators (DRs), which are also referred to as morphogenic regulators (MRs), such as the WUS (WUSCHEL), BBM (BABY BOOM) and GRFs (GROWTH-REGULATING FACTORs), have been proven to overcome genotype dependence in plant regeneration [6,7,8,9,10,11], while in many plant species, the delivery of exogenous DNA or RNA is genotype dependent and is the main bottleneck in genetic transformation. Therefore, there is an urgent need to develop new delivery vectors for genome engineering.

The clustered regularly interspaced short palindromic repeats (crispr)/crispr associated protein (Cas) genome editing system is the most popular and promising genome engineering tool in the past decade, because of its ability to precisely modify target DNA with high efficiency and conveniency [12]. This RNA-guided endonuclease system can be delivered into cells in the form of plasmid DNA, DNA fragments, RNA, and ribonucleoproteins (RNPs) [13]; other materials can only be conveyed in certain forms, such as single-stranded oligo donor (ssODN) and double-stranded oligo donor (dsODN) that are commonly used in target fragment insertions and replacements [14,15]. As compared with studies in mammalian cells and bacteria, which could directly escort these materials into single cells with well-established lipofectamine and polyethyleneimine (PEI) transformation assay, the presence of the cell wall and the requirements of the tissue culture process make delivery of these materials quite challenging in plants. Some of these materials are quite critical, such as the RNA and RNPs, which can generate transgene-free genome-edited plants without the integration of exogenous DNA sequences [13]. These generated plants are not regarded as genetically modified organisms (GMOs) in many countries [16]. Thus, new delivery methods with various cargo types and high efficiency are key to applying genome editing in crop breeding.

A special lecture entitled “There’s plenty room at the bottom” expanded the influence of nanomaterials [17]. In the 21st century, there have been rapid developments in nanomaterials which are used in some of the most advanced technologies and have the potential to revolutionize fields such as catalytic science, electronic technology, medicine, and agriculture [18]. Due to their small size (<100 nm), nanomaterials exhibit unique physical and chemical properties, and further potential applications in gene delivery have been taken into consideration. For example, several nanomaterials have been reported to be directly taken up by animal and plant cells. Nanomaterials have been commonly used as drug vectors in animal cells. Modifications can be made to nanomaterial surfaces that allow them to act as vectors to deliver nucleotides and proteins. Inspired using transgenic nanomaterials in animals and human cells, some studies have reported that nanomaterials allow foreign plasmid DNA or dsRNA to pass through rigid plant cell walls and remain functional. In this review, we summarize the advantages and disadvantages of traditional gene delivery methods. Furthermore, we focus on the prospects of nanomaterials for future crop breeding and genome engineering in plants.

## 2. The Merits and Limitations of Traditional Delivery Methods

Genome engineering is an important strategy for sustainable crop development and breeding and it is expected to be the key in future breeding efforts [19]. Commonly used methods of engineering in the last century have included Agrobacterium-mediated infection, particle bombardment, PEG-induced transfection, and viral transfection showed as the Table 1 [20,21]. 

Agrobacterium-mediated infection is the most widely used delivery system in plant genetic engineering [22]. Transfer DNA (T-DNA) is released from the binary vectors, which is engineered from the Agrobacterium tumor-inducing (Ti) plasmid and subsequently integrated into the plant genome [23]. As dicots are the natural hosts for Agrobacterium, Agrobacterium-mediated infection is relatively easier in dicots as compared with monocots. This is mainly due to the fact that wounds cause phenolic signal molecules in dicot plants to induce the expression of Vir genes to facilitate the transformation process, which is absent in monocots. Adding exogenous acetosyringone (AS) can also activate the expression of the Vir genes, thus, it is required for Agrobacterium-mediated transfection of monocot plants. This obstacle can also be bypassed by constitutive strains, which constitutively activate the expression of Vir genes [24,25]. Now, in many model monocot varieties, Agrobacterium-mediated transformation is quite efficient and stable. However, in many other cultivars, both in monocots and dicots, Agrobacterium-mediated genetic transformation is still quite challenging.

There are many factors that affect the efficiency of Agrobacterium-mediated transformation [26]. The most well-known barrier is the genotype dependence among different varieties even from the same plant species. Differences in genetic background, and the physical and metabolic conditions lead to different responses to Agrobacterium infection [27,28,29]. Different regeneration abilities among cultivars have been observed almost in all plants. DRs are activated during plant regeneration, with dramatic responses in different plant species. In recent years, transient or stable overexpression of DRs, such as BBM, WUS, GRF5, and GRF4-GIF1, have been proved to successfully overcome this bottleneck in many plants’ genetic transformation processes [6,7,8,9,10]. 

However, in some plant species, the physical and metabolic properties make Agrobacterium an unsuitable vector to deliver exogenous DNAs [26]. For example, the specialized metabolites (such as ethylene, salicylic acid, and γ-aminobutyric acid (GABA)) in many varieties inhibit the growth and the gene transferring efficiency of Agrobacterium [30,31,32]. Antagonism between some host plants and Agrobacterium can lead to a diseased state [33]. Furthermore, tissue culture conditions, such as the density of Agrobacterium, the concentration of AS, the preculture course, the co-inoculation time, and the types and concentrations of hormones need to be tested for every variety. Apart from this, other transformation materials, such as RNA, RNPs, or oligonucleotides, cannot be delivered through Agrobacterium. In addition, these materials are quite important in the genome editing of crops to produce transgene-free and precise genome-edited varieties [34,35]. Thus, Agrobacterium-mediated transformation is simple and convenient but it is not suitable for all plant species or purposes, and its efficiency should be improved. 

Particle bombardment is carried out by a biolistic delivery system, also known as a gene gun, which was established in 1987. Exogenous biomolecules (such as DNA, RNA, proteins, and peptides) bound to gold particles pass through the plant cell wall to reach the nucleus and other organelles with the help of external high pressure [36,37,38]. This method is considered to be one of the most promising methods in plant genetic engineering, because there is no limit in the genetic materials and the tissue types used, especially for transformation of mitochondria and chloroplasts [39]. Nevertheless, its application in commercial breeding is quite limited due to random and multiple-copy insertions of target DNA, and the often observed chromosome deletions, translocations, and inversions [40], which are the result of highly pressurized particles [41].

PEG-induced protoplast transfection was the earliest method used in cereal crop transformation. PEG solution reverses the permeability of the cell membrane, which allows foreign genes to easily pass into the nucleus [42]. This method is highly efficient, relatively easy to perform on many plant species, and has been used to deliver plasmids, RNA, and RNPs [13,43]. Thus, it has been widely used to transiently investigate the localization and functions of desired genes and to identify the genome editing efficiency of target sites. However, regeneration of transfected protoplasts into mature plants is time-consuming, laborious, and has highly technical requirements [44]. 

Plant viruses can also be used to deliver genes; genes of interest can be packed and replicated during the viral replication cycle [45]. Viral vectors have been widely used to study gene function via the virus-induced gene silencing (VIGS) system [46]. Last year, a method was reported in which Agrobacterium with virus vectors was sprayed on leaves to improve crop features [47]. Recently, viral vectors have been used to deliver single guide RNAs (sgRNAs) to transgenic cotton, tobacco, Arabidopsis thaliana, tomato, wheat, and maize plants overexpressing Cas9. Formation of the Cas9–sgRNA complex allowed for precise genome editing at the target site. Genome-edited plants that were free of viruses could be obtained in the next generation [48]. However, many concerns existed in virus vectors. Utilization of plant viruses is a great safety concern in plant breeding, as many viruses, seed borne and transmissible, cause severe growth defects. Engineered virus vectors undermine the toxicity of these viruses, but they can still cause growth abnormities. Endogenous gene silencing of viruses is a major resistance for the efficiency of virus vectors, which explains why tobacco is the most used host for virus and virus vector studies, as the gene silencing of viruses is very weak in tobacco [49]. The host range of viruses also limits their application. The tobacco rattle virus (TRV) vectors are the most widely used virus vectors in dicot and monocot plants, but they cannot be used in nonhost plants such as rice [50]. The virus vectors need to be systemically spread, and the spread time and operation requires high skills.
ijms-23-08501-t001_Table 1Table 1The merits and limitations of traditional delivery methods of plants genome engineering.Traditional Delivery MethodsMeritsLimitationsCargo TypesRef.*Agrobacterium*Well-established protocols, low cost and widely usedGenotype-dependent;limited cargo type; Antagonism between *Agrobacterium* and plants, not appliable in several plant species; cargo type limitationPlasmid DNA[22,26]Particle bombardmentPromising in the genome engineering of mitochondria and chloroplasts, suitable for all cargosRandom insertions, tissue type depended, host genome damages often happen, expensive equipment and materialsPlasmid DNA, RNA, RNPs, synthesized oligonucleotide[37,38,41]PEGHighly efficient in protoplast, suitable for all cargosTime-consuming, cell limitations, regeneration inefficient, polyploid formationPlasmid DNA, RNA, RNPs, synthesized oligonucleotide[13,42]Plant virusGenotype-independent, high level of transient expressionCargo size limitations,plant species restricted,safety concern in crop yieldDNA, RNA[46,48]


## 3. Nanomaterial-Induced Gene Delivery Systems

Nanoscale materials exhibit a significant advantage in agricultural activities [51]. Nanonutrients, -pesticides, and -fertilizers promote plant health and yield better than common commercial formulations with equal or lower concentrations, smaller size which facilitates their better physical and chemical performance, and they can be easily assimilated by cells. Nanopesticides and -fertilizers are nearly 30% more efficient than conventional applications [52]. Furthermore, nanomaterials have minimal toxicity, which may yield more opportunities for future agricultural development [53]. 

The unique properties of nanomaterials make them easily obtained by plant cells. Nanoparticles such as pesticides and fertilizers have been confirmed via transmission electron microscopy to be delivered into plant cells. However, the mechanisms by which such nanoparticles enter plant cells have only been partially uncovered. A lipid exchange envelope penetration (LEEP) model was proposed to explain how nanotubes penetrate the double lipid layer of plant cells [54]. Traditionally, cuticular and stomatal uptake have been considered to be the two main pathways of nanomaterial entry into foliar and cells [55]. The specific micro/nanostructures on leaf surfaces in different plant species affect the entry of nanomaterials into leaves [56]. Particle size is the primary factor limiting the entry of nanoparticles into plants. Yong et al. (2021) reasoned that 50 nm was the baseline for nanomaterials that could freely penetrate plant pollen cells [57]. Other studies have suggested that the size of nanoparticles that can freely enter plant cells should be less than 20 nm at least in one dimension. Other nanomaterial features (e.g., shape and zeta potential) also affect entry into plant cells. Nano-delivery systems have high transferring efficiency when the net zeta potential values are over 30 mV, because a low zeta potential value can induce aggregation and a high value makes the delivery system more stable [58,59]. When the nano-plasmid complex is injected into or absorbed by plants, it can be released, and high transferring efficiency is obtained [58]. Zhang et al. used different shapes and particle sizes of AuNPs (sphere, bar) to inject RNAi into GFP-overexpressing *N*. *benthamiana* leaves and found the bar AuNPs had higher transferring efficiency [60].

Nanomaterials have successfully been used to mediate foreign gene expression and CRISPR/Cas-based genome editing in human cells to treat diseases [61]. The cell wall is a natural barrier that blocks entry of foreign materials into the cytoplasm or organelles of plants [62]. Recently, various nanomaterials have been reported as vectors for delivering foreign genes into plant cells, such as mesoporous silica nanoparticles (MSNs), liposome, modified magnetic metal particles, layered double hydroxides, and carbon tubes; for each of these materials, at least one dimension is smaller than 100 nm. Two methods have been described for loading plasmids onto nanoparticles: ion exchange (IE) and electrostatic adsorption. As compared with IE, the electrostatic adsorption method adsorbs nearly 100% of the plasmid within a short time period and gives strong protection from deoxyribonuclease I (Figure 1). 

It was first reported in 1992 that a micromaterial, silicon carbide fibers, was utilized to deliver plasmid with GUS and Bar genes into tobacco suspension cells [63]. A Southern blot analysis proved a high percentage of positive transformation events. They proposed that silicon carbide fibers, which are like a needle, hooked on the tissues and penetrated into the cells. This method was applied to wheat and maize mature embryos, and transgenic maize was harvested. Strong GUS expression was observed in offspring after two years [64]. In 2008, salt tolerant cottons were reported by using silicon carbide whiskers to introduce a foreign gene [65]. The exact mechanism for this method is still in dispute. It is possible that the silicon carbide whiskers do not carry the gene, but they are hard enough to penetrate cell walls and open access for gene to enter the plant callus (cotton, maize, and wheat). Since then, gene nanovector technology has been gradually developed and reported.

Due to the intrinsic hardness and excellent shock of silicon nanomaterials, lots of decorated MSNs surfaces are used as vectors, which has drawn a great deal of attention in recent years. MSN particles have been used to deliver pesticides, fertilizers, and other agricultural chemicals [66,67]. Torney et al. (2007) used different functional groups to decorate MSNs. These functional MSNs with green fluorescence label can penetrate tobacco protoplast without external force. Additionally, decorated MSNs have been used as the ‘powder’ of the gene gun to replace the expensive gold powder required in bombardment of leaves and callus, and the GFP signal can be detected within a short time interval [68]. MSNs decorated with gold nanoparticles (Au-MSNs) have been used to co-deliver plasmids and proteins into white onion epidermis cells with the help of a gene gun, and the green signal was detected after 24 h [69]. The Au-MSNs release plasmids and proteins after passing through cell walls. To find other routines for the nano-mediated transgene, smaller MSNs (~40 nm) bound to DNA can be absorbed by tomato leaf and shoot cells after spraying or injection. The GUS gene was applied to test the feasibility of the delivery system. Reverse transcription (RT)-PCR and Western blot were conducted to monitor gene expression and demonstrated that materials less than 100 nm in size could pass through the cell wall without assistance depicted as the Figure 2 [70].

There have been reports in which magnetic particles are used as the vector to enter plant cells through larger pores such as pollen cells or protoplasts, and foreign genes are then expressed with the help of a magnetic field. FITC signals, which are delivered by PEG-modified magnetic gold particles (mGNPs, ~30 nm) have been observed in over 95% of canola protoplasts [71]. Plasmid with GUS gene bound with mGNPs were transferred into canola cells and GUS expression was detected after co-culturing for 48 h. In 2017, magnetic Fe_3_O_4_ was decorated with branched polyethyleneimine (PEI) and introduced into pollen cells via the pollen aperture. This system was tested by GUS gene in many plant species including cotton, lily, chili pepper, etc. Furthermore, Bt-resistance genes have been transferred in to cotton pollens. Transgenic plants were obtained and it was hypothesized that integration of the foreign genes into the cotton genome was responsible for the high pest resistance observed in the offspring. This method was much faster than the traditional breeding process [72]. Similar methods have also been reported in maize. It has been found that pollen can absorb DNA nanoparticles with high ability under specific conditions; maize embryos have been confirmed to contain GUS genes transferred by magnetic nanoparticles [73]. Similarly, the offspring of transgenic maize displayed the desired features of foreign genes vividly showed as the Figure 3.

Single and multiple wall nanotubes (SWNTs and MWNTs, respectively) have both been used as gene vectors in plants. Demirer et al. (2019) used positively charged materials including PEI (with different molecular weights), chitosan, and sodium dodecyl sulfate (SDS) as the nucleic acid joint to modify SWNTs and MWNTs. In addition, green fluorescent protein (GFP) and yellow fluorescent protein (YFP) have been successfully expressed in *N. benthamiana* leaves and even in the chloroplast using decorated SWNTs as the vector after injection or co-culture [59,74]. They found this delivery system coulf be applied to wheat, cotton, and arugula. However, GFP signals diminished within one week, which may be due to physical damage caused by injection of foreign genes. Furthermore, GFP signals have been observed in protoplasts several hours after penetration by SWNT with SDS decorating. This technique was used to knock down target genes by delivering siRNA into GFP-overexpressing tobacco plants just as the Figure 2a,b [75]. A series of studies by the same authors clearly demonstrated the potential of nanomaterials in gene engineering assays and applied this technique as a sensor for plant disease detection [76].

Other nanomaterials have also received attention as potential gene vectors. Layered double hydroxides (LDHs) is a 2D nano-sheet material that can easily be synthesized and contains highly positive charges. As compared with other methods, it is more compatible with animal cells (including human cells). LDHs show better leaf adhesion as a result of multiple hydroxyl groups among the layers [57]. Traditionally, LDHs have been used as a tracker for medicine and agrichemicals [77,78]. Bao et al. (2016) used LDHs to introduce an FITC–DNA short fragment into BY2 and *Arabidopsis* root cells, and strong signals of FITC were detected within 2 days [79]. Double-stranded RNA has shown great potential to replace genetically modified crops for pest control in the future [80]. Mitter et al. (2017) used LDH nano-sheets as a dsRNA vector, which gave plants long-lasting (over 20 days) protection against viruses as compared with a direct spray of dsRNA and the RNA obtained better protection due to nanovectors. Double standed RNA was hard to wash off plant leaves when it was loaded on LDHs [81]. Liu et al. (2020) obtained similar results using LDHs; plants showed better resistance to whitefly-transmitted begomoviruses when LDHs were used as a vector of artificial microRNA [82]. As compared with other nanomaterials, LDHs are safer and can be decomposed in ambient air. Therefore, they may be a promising nanovector for genome engineering. The relatively small LDH–RNA complexes can also enter plant pollen without external force and show higher knockdown efficiency than larger complexes. Yong et al. synthesized four classes of LDHs (nearly 120, 80, 50 and 30 nm). They demonstrated that, when the nanovector particle size was smaller than 50 nm, it could penetrate into plant cells freely. The smallest vectors (30 nm) showed the best silencing efficiency to target gene [58]. The unique adhesive properties of plant leaves may allow for novel methods of LDH application. The above results indicate that LDH is an advanced gene vector for plants.

The smallest nanomaterials (less than 20 nm), such as carbon dots (CDs) and AuNPs, also show great promise as gene vectors in plants. Crops treated with CDs show enhanced yield and great resistance to adversity [83]. A CD–DNA complex applied with a surfactant resulted in GFP expression in mature tomato leaves. Then, they realized target gene knockdown in GFP overexpression plants [84]. Huang et al. used PEI-decorated CDs, whose particle size was less than 10 nm, as the vector realized foreign gene expression. Firstly, they used the CDs to take the report systems (mcherry and GFP) realized expression in rice root and wheat leaves. GUS gene expression was delivered by PEI-CDs in fresh rice callus under the help of vacuum infiltration. The nanovector was used to transfer the plasmid expressing the hygromycin resistance gene into wheat leaves which showed great resistance in the hygromycin solution as compared with the wide type. [85]. AuNPs have shown advanced applications in unique optical, sensor, biochemical, etc. They are also used as the gene vector for animal cells. This delivery system has minimal cytotoxicity [86]. However, the use of this vector in plant cells has rarely been reported. Zhang et al. 2021) used the different sizes and shapes of AuNPs to take DNA-Cy3 into plant cells by injection [60]. Rod-shaped AuNPs can internalize into plant cells but the 10 nm spherical AuNPs show high RNAi efficiency to target gene [60].

Actually, the other organic polymers and liposome are often used in genome engineering works, especially in animal and clinical treatment of diseases [87,88]. In 2021, liposomes as the vector of the CRISPR-Cas system were applied for human cancer therapy [61]. Liposomes are synthesized by different single amphiphilic lipids or different lipids which are charged or neutral [89]. Liposomes have better compatibility to biomolecules including proteins, enzymes, and nucleic acids [90]. The spraying method has been used to apply liposomes to plant leaves by Karny et al. They found the active ingredient (contained Mg and Ca) can be realized in plants [91].

Because of the special characteristic of liposomes, they are often used as the carrier for genes into cells by infusion with cytomembrane or endocytosis [92]. Wang et al. (1986) used calcein fluorescence as the marker to detect the transferring rate in carrot protoplasts, over 89% of which could absorb negative liposomes [93]. Liposomes can envelope various materials including DNA plasmid, RNA, and different sized DNA fragments. Tobacco mosaic virus (TMV) RNA has been entrapped in negative liposomes, which could be expressed in target cells. According to previous results, entrapped TMV RNA infected Vinca, Petunia, and the protoplasts of tobacco [94,95,96,97,98]. GAD et al. (1988) found that negative liposomes could adhere to watermelon pollen grains and other organs [99]. Two works have disclosed that positive charged liposomes showed much lower efficiency (almost 0.05%) [98,100]. In addition to protoplast, liposome can also enter the pollen cells and callus. It was reported by GAD et al. that 150 nm liposomes helped macromolecular material enter budding watermelon pollen [99]. At the same time, a positive charged liposomes have been reported to be used to deliver labeled DNA into germinating pea pollen grains [101]. Tissue culture technology is often used during genome engineering works. Rosenberg et al. (1988) used liposomes to deliver the foreign DNA of chloramphenicol acetyltransferase and tomato yellow leaf curl virus into tomato and tobacco callus. The DNA expression was demonstrated by Southern blot [102,103]. Based on the above research, liposom can take DNA into protoplast and also can reach specific tissues. Until now, there have not been any works published that reported liposomes could pass through cell walls and obtain genetic transformations.

Other natural high-molecular polymers such as chitosan, starch can be used as the vector or ligand between nucleic acid and nanoparticles. Chitosan has applied SWNT and delivered plasmid into chloroplast and realized expression [60]. The super features of chitosan have broad prospect in organelle genome engineering.

Liu et al. (2008) synthesized positive charged starch sphere (less than 100 nm) as the vehicle for plant genetic transformation. Plasmid was loaded on the surface of spheres and the complex entered the suspension cells treated with ultrasound. However, the ultrasound often induced cell damage. There is a great need to search for new routines for polymer nanovectors [104].

The shapes of various nanomaterials even the DNA-nanostrucutre can promote movement of exogenous DNA or RNA into some plants or explant cells either with or without further assistance [105]. Agrobacteria can directly or indirectly introduce DNA and RNA into explants to alter the host genome [106]. However, it is difficult to take the RNP, short or modified, donor to reach efficient transgene-free and homology directed repair (HDR) for genome engineering in plants. Nanomaterials have small particle sizes and large surface areas, which provide anchor spaces for different sizes of plasmids or DNA fragments [107]. Gene guns can be used to transform cells with a mixture of gold powder and DNA, RNA, or RNPs. However, the strong pulses required may fragment the host genome, and random insertions commonly occur. It takes a great deal of time and it is difficult to obtain homozygotes. In contrast, nanovectors can introduce foreign genes into cells in a manner similar to nutrient absorption, decreasing potential harm to the host genome. Nanomaterials has low toxicity to plants. Inspired by the nano-delivery systems used for other organic materials, nanovectors could be developed as foreign gene vectors for advanced genetic engineering methods for different organelle of plants in future breeding works.

## 4. Future Prospects for Plant Genetic Engineering with Nanomaterials

Previous studies described above have demonstrated the great value and potential of nanomaterials for gene delivery showed in Table 2. However, many factors and the detailed mechanisms associated with these methods remain to be resolved.

It is still unclear whether foreign DNA delivered into plant cells by nanomaterials is integrated into the plant genome. Nanovectors lack genes that are essential for the integration of T-DNAs, such as the Vir genes from the Ti plasmid. Foreign DNA has been shown to be quickly degraded during cell replication in plant cells [59]. However, studies have shown that foreign DNA delivered into cotton and maize pollen cells using magnetic nanomaterials indeed integrated into the genome. Moreover, the expression of reporter genes, such as GUS and GFP, have been detected in progeny seedlings [72,73], demonstrating the stable integration and inheritance of exogenous DNA. However, detailed analyses, such as the identification of the integration sites and the border sequences of the integrated foreign DNA, remain to be conducted. The underlying mechanisms of these integration processes are key research topics in the future.

Successfully transient expression of cargo DNAs have been achieved with several nanomaterials in plants. However, the expression efficiency may not be enough for genetic engineering, especially genome editing. In the work of Ali et al. (2022), they used SWNT as the vector, which was similar to Demirer et al. (2019), to introduce foreign genes into mature plant leaves [108]. As compared with a traditional Agrobacterium-mediated delivery system, very weak signals were detected surrounding the infiltration sites when nanomaterials were used to deliver plasmids containing GFP. When cotton was injected by the complex of the DNA and nanovector, low level expression was detected by confocal microscopy. This was mainly due to the transient expression of target DNAs, instead of stable expression, and the uptake efficiency of cargo plasmids [108]. The CRISPR-Cas system is a promising method for genome editing in plants to enhance crop yield, adversity resistance, and disease resistance. However, the efficiency is far from satisfactory even with the traditional delivery method [109]. To ensure the efficient and successful editing of desired sites, strong, constitutive, and sometimes virus promoters have often been used to boost the expression level of Cas and sgRNA [110,111,112,113]. These methods could be used in nanomaterial-mediated genome editing, and also in other nanomaterial-based plant genetic engineering. The types of nanomaterials, modifications, the size and type of the DNA vector, and the production and transfection processes, abovementioned, play significant roles in determining delivery efficiency [60]. We could also elevate expression efficiency by increasing the number of exogenous genes. At present, delivery efficiency is highly variable between labs. We speculate that the complex synthesized process, explant types, and status may affect the transformation efficiency. All these factors have limited widespread use among different labs. It is, therefore, urgently needed to establish a simple, stable, and highly efficient nano-induced delivery system.

The cytotoxicity of nanomaterials is a major concern to further agricultural activity applications. As advanced gene vectors for plants, all the reported materials have shown non-cytotoxicity to plant cells or organs. Various nanomaterials can help plants or crops gain more resistance to adversities or gain yield. As we know, PEI is used as the ligand for DNA, siRNA, and miRNA because of its positive charge. Traditionally, the high molecule weight of PEI has more toxicity to animal cells [114]. To date, all the representative nanovectors show safety to plants or protoplast. To avoid the occurrence of cytotoxicity, natural original or low-toxicity positive compounds such as chitosan, arginine, and low molecule weight PEI are recommended [74,115].

Nanomaterials are promising for gene delivery in plants because they can overcome species limitations in genetic transformation with conventional method in cereals, vegetables, and fruits. As a result, many labs have attempted to use nanomaterials for the delivery of genome editing systems, which are promising for crop breeding. In contrast with Agrobacterium-mediated methods, DNA as well as RNA and RNPs can be delivered via nanomaterials showed in Figure 4. Plant scientists have attempted to deliver ssODNs and dsODNs as the template for HDR or as the cargo for target insertion and replacement [116], which has been conducted in mammals. With traditional delivery systems, this is mainly accomplished through particle bombardment, which often generates multiple insertions of oligodeoxynucleotides and the Cas9 gene fragments [15], making this approach unsuitable for commercial use. These difficulties could be solved using nanomaterials.

New selection strategies must be established for nanomaterial-based genome engineering as compared with traditional integration-based gene delivery systems. Transgenic or gene-edited plants via traditional genetic engineering methods are often screened with selection markers (e.g., kanamycin, hygromycin, or spectinomycin). However, the integration of foreign genes with nanovector is still unclear and delivery efficiency is currently limited. It is well known that gene editing is very important for future breeding works, and it may be achieved during the transient expression processes. There is a great possibility to realize gene editing affairs without insertion [117]. The new method may give strong support to obtain transgene free plants [58]. There is an urgent need to develop efficient screening methods for genetically engineered plants using more sensitive selection markers. Furthermore, new transformation routines should also be developed based on nanomaterial induction and shortened tissue-culture cycles [118].

## 5. Conclusions

Genetic engineering is an efficient way to obtain specific agricultural characteristics in crops. The CRISPR-Cas system uses highly specific nucleases to create double-strand breaks (DSBs) in target sites; this approach has a promising future in crop breeding. Many editing affairs are caused by transient expression in a short time [110]. As we all known, the strict regulation of GMOs has promoted the development of plant genetic engineering techniques in which target or functional gene expression is instantaneous and foreign DNA is not integrated into the genome of host plants. However, technical barriers have emerged in genetic engineering and are difficult to overcome by conventional methods such as the high random insertion ratios of gene guns. Furthermore, bombardment pressure may cause host genome damage and active multiple repairing mechanisms. Agrobacterium infection is commonly used for plant genetic works. It also can induce random insertions and it is hard to obtain transgene-free plants. The PEG method is a special approach for transgene works because the protoplast is harvested from leaves without cell walls. The various foreign gene cargos can pass the lipid bilayer and reach the nucleus or organelles. However, the differentiation from protoplasts into plants is a time-consuming and challenging task. Functional nanomaterials show superior ability to deliver a variety of sizes of DNA, RNA, and RNPs in animal cells [119,120,121]. They exhibit great potential for relatively simple and highly efficient HDR or transgene-free engineering in plants. It is necessary to design vectors and enrolled exogenous genes to enhance differentiation or regeneration. To sum up, the nanomaterial delivery system is more bio-compatible, non-toxicity, and low cost as compared with conventional methods. Nanovectors have exciting potential for advanced applications in future crop breeding.

## Figures and Tables

**Figure 1 ijms-23-08501-f001:**
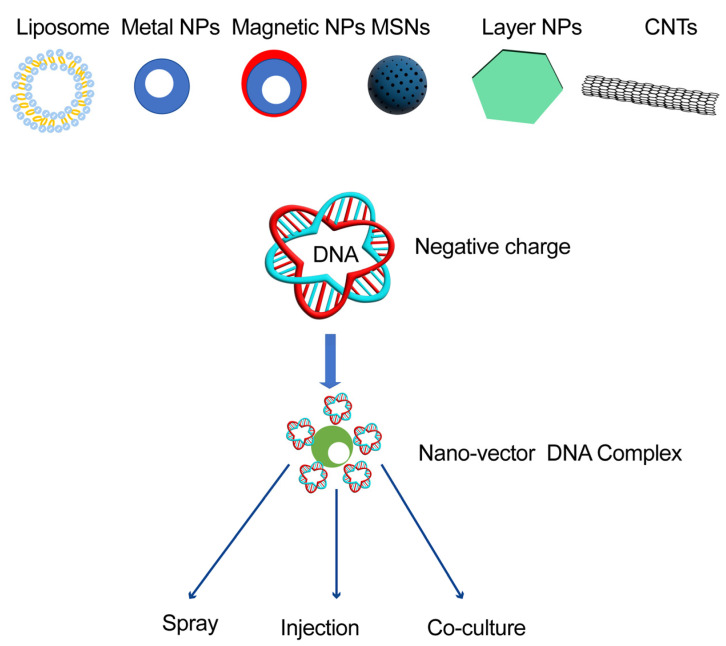
The gene nanovectors for future breeding works in different plants. Several types of nanomaterials, such as the liposome, modified metal nanoparticles (NPs), magnetic NPs, mesoporous silica nanoparticles (MSNs), layered double hydroxides (layer NPs), and carbon nanotubes (CNTs) have been used to deliver cargo genes into plants. Since the target nucleotide (DNA and RNA) are negatively charged, these nanovectors are positively charged. The nanomaterial–DNA complex could be delivered to plant cells by spraying, injection, or co-culturing, to improve the performance of the plants.

**Figure 2 ijms-23-08501-f002:**
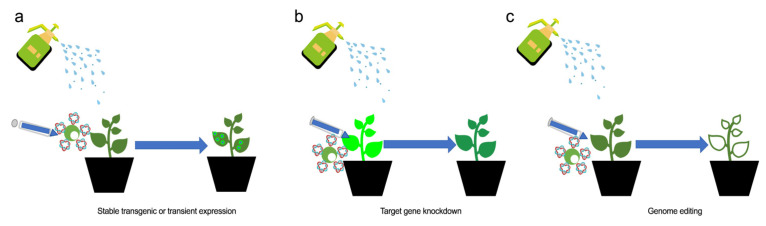
The application of nanomaterials in plant genetic engineering. By spraying, injection, and co-culturing cargo-packaged nanomaterials, plants can be engineered using stable integration or transient expression of exogenous genes (**a**), or by knockdown of target genes by delivering microRNAs (**b**), or by precise genome editing through the delivery of the CRISPR/Cas system (**c**).

**Figure 3 ijms-23-08501-f003:**
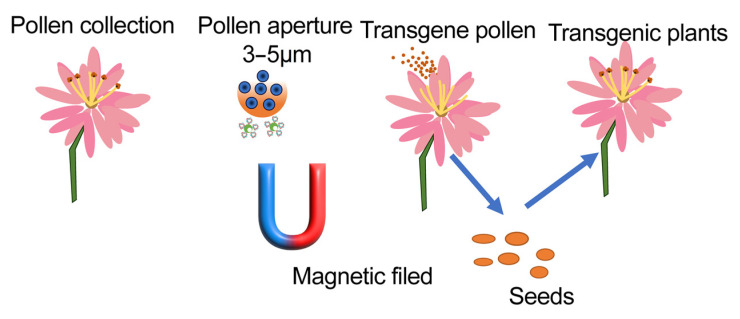
The magnetic nanovectors induce genome engineering with pollens of plants. Plant pollens (about 3–5 μm) are collected and incubated with nanomaterial–DNA complexes in a magnetic field. Then, the transgene DNA-containing pollens are sprayed onto the surface of pistil stigma to obtain the transgenic seeds.

**Figure 4 ijms-23-08501-f004:**
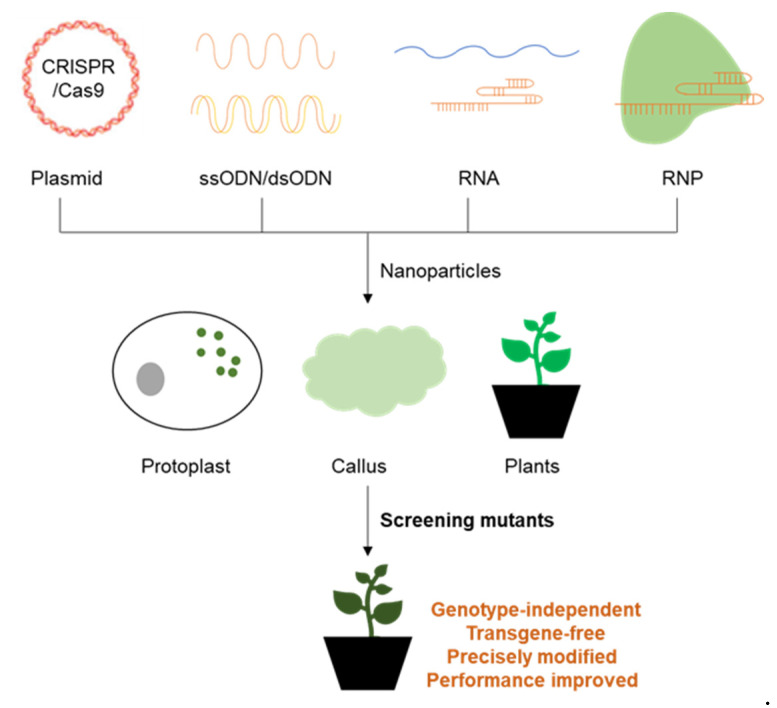
The promising application of nanomaterials in plant genome editing. The ability of nanomaterials in delivering all types of cargos, including plasmids encoding the CRISPR/Cas system, single-strand and double-strand oligo donor (ssODN and dsODN), RNA and ribonucleoproteins (RNPs), makes them quite promising in plant genome editing, since traditional delivery methods can only deliver certain types of cargos or have other drawbacks. These nanomaterials could be directly used to transform protoplasts, callus, and plants, and precisely edit the genome. New selection and transformation methods are needed to screen and finally obtain the mutant plants. These methods are genotype independent, we can acquire new plant varieties without integration of exogenous DNA, and the genome can be precisely modified as we wish, to achieve better performances in the field and ensure food safety.

**Table 2 ijms-23-08501-t002:** Summary of the micro/nanovectors for plants in recent years.

Materials-Vectors	Cargos	Plants	Cell Types	Delivery Methods	Ref.
Silicon carbide fibers	Plasmid (contains Bar and GUS)	*N. tabacum*, maize, rice, ryegrass, and cotton	Cells suspension, callus	Co-culture	[63,64,65]
Gold functional MSNs	Plasmid (GFP gene)	*N. tabacum*, maize, white onion	Mesophyll protoplasts, epidermis cells	Gene guns	[68,70]
Functional MSNs	Plasmid (GUS gene)	Tomato	Epidermis cells	Spraying or injection	[69]
Magnetic NPs	Plasmid (GFP, GUS and Bt gene)	Cotton, lily, maize	Pollen cells	Magnetic field	[72,73]
Layer double hydroxides	dsRNA for RNAi	Cowpea, *A. thaliana*, *N*. *tabacum*, *N. benthamiana,* *S. lycopersicum,*	Mature leaves	Spraying	[81,82]
		tomato	Pollen cells	Co-culture	[54]
SWNT/MWNT	Plasmid (GFP, YFP) siRNA for RNAi	*N. benthamiana, E. sativa,* arugula, *A. thaliana*	Mature leaves, protoplast and chloroplast	injection without needle or co-culture	[58,74,75]
Carbon dots	Plasmids (GFP, GUS, hygromycin resistance gene); siRNA for RNAi	Wheat, rice, tomato	Mature leaves, callus	Spraying/ vacuum assisted	[84,85]
Different shapes of AuNPs or magnetic	Plasmid (GFP); siRNA for RNAi	*N. benthamiana*	Mature leaves	Injection without needle	[60,71]
DNA-nanostructure	siRNA for RNAi	*N. benthamiana*	Mature leaves	Injection without needle	[105]
Liposomes	Plasmid of DNA (acetyl transferase) TMV-RNA	Watermelon, Tobacco, *Vinca, Petunia, Pea*	Pollen cells Protoplast	Co-culture	[91,92,93,94,95,96,97,98,99,100,101,102,103]

## Data Availability

Not applicable.

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
