# Peer review of "The Promising Nanovectors for Gene Delivery in Plant Genome Engineering"

_ijms, 2022, doi:10.3390/ijms23158501_

Round 1

Reviewer 1 Report

To actualizing genetic engineering in plants and crops by highly efficient gene delivery system, a couple of viral and non-viral transfection methodologies were implemented to deliver gene substances to many crops/plant species. Herein, the authors summarized recent progress of non-viral nanomaterials for plant gene transfection and discussed the advantages and limitations of different transfection methods, moreover, they emphasized the advantages and potential broad applications of nanomaterials in plant genome editing. This article provided useful information of non-viral nanomaterials in plant genetic engineering and crop breeding. There are some issues need to be addressed before further consideration:

1.      The abstract is not well-organized, the authors need to state the most important discovery of this field briefly.

2.      There are some lexical errors in the article, e.g., page 4, table 2, line 3, “radom insertions” should be “random insertions”; page 5, paragraph 4, line9, “antoher" should be “another”. The authors please carefully proofread through the text.

3.      The authors mentioned that “the net zeta potential above 30 mV which could make delivery system more stable”. In general, the increased zeta potential could increase the gene transfection efficiency but there is no direct correlation between the stability of delivery system and zeta potential, the authors please clarify this part.

4.      The physico-chemical factors such as: particle sizes, zeta potential, nanoparticle morphology, as well as chemical components that greatly affected the gene transfection efficiency. The authors should discuss more about these physico-chemical factors in the main text.

5.      For biological features of the non-viral plant gene transfection nanomaterials such as: cytotoxicity, gene (DNA, siRNA, miRNA) transfection efficiency, cellular uptake efficiency, as well as the cellular/tissue distribution need to be deeply discussed.

6.      In animal cell-based gene transfection, lipofectamine and Polyethyleneimine (PEI) were usually selected as the “gold standards”, is there any materials set as “gold standards” in plant cell gene transfection system?

7.      The conclusion part is a bit simple; the advantages and disadvantages of current plant gene delivery system should be clearly presented. Moreover, the future perspective in this field needs to be discussed. 

Author Response

To actualizing genetic engineering in plants and crops by highly efficient gene delivery system, a couple of viral and non-viral transfection methodologies were implemented to deliver gene substances to many crops/plant species. Herein, the authors summarized recent progress of non-viral nanomaterials for plant gene transfection and discussed the advantages and limitations of different transfection methods, moreover, they emphasized the advantages and potential broad applications of nanomaterials in plant genome editing. This article provided useful information of non-viral nanomaterials in plant genetic engineering and crop breeding. There are some issues need to be addressed before further consideration:

  1. The abstract is not well-organized, the authors need to state the most important discovery of this field

> Thanks for your constructive suggestions. We furtherly organized the abstract in this version. We have added the important discovery of the application of nanomaterials in plants, such as the successful delivery of cargo genes into plants and the generation of genome edited cotton and maize plants. Please refer to line 10-25.

  1. There are some lexical errors in the article, e.g., page 4, table 2, line 3, “radom insertions” should be “random insertions”; page 5, paragraph 4, line9, “antoher" should be “another”. The authors please carefully proofread through the text.

> Thanks for your nice advice. We have carefully checked the manuscript again and again, and corrected all the spelling mistakes.

  1. The authors mentioned that “the net zeta potential above 30 mV which could make delivery system more stable”. In general, the increased zeta potential could increase the gene transfection efficiency but there is no direct correlation between the stability of delivery system and zeta potential, the authors please clarify this part.

> Thanks for your kindly questions. As we all known the colloid system could show high stable in environment atmosphere. The high zeta potential showed better transfer efficiency. And according to the results of Ali, which they used different mass ratio between the DNA and nano tubes, the high ratio of the DNA/nanotube poorly showed stability. And another work published by the Demirer et. al they also showed the high zeta potential could help the delivery system to keep stable, as we can see from the picture below; the low zeta potential value may induce aggregation. Furtherly, in the book chapter Multifunctional nanocrystals for cancer therapy: a potential nanocarrier, Joseph et al., illustrated the net value over 30 mv is generally considered to have sufficient repulsive force to attain better physical colloidal stability. We proposed the stable delivery system could reach the high transient expression. We have modified our statements and added these references, please refer to line 191-194.

  1. The physico-chemical factors such as: particle sizes, zeta potential, nanoparticle morphology, as well as chemical components that greatly affected the gene transfection efficiency. The authors should discuss more about these physico-chemical factors in the main text.

> Thanks for your nice advice. We added the details discuss about how the physic-chemical factors affect the transfections in this version. Page 5, line 185-199 page 9, 366-380. We discussed the particle size, zeta potential and the shapes of the nano-vector how to influence the gene transfection efficiency.

  1. For biological features of the non-viral plant gene transfection nanomaterials such as: cytotoxicity, gene (DNA, siRNA, miRNA) transfection efficiency, cellular uptake efficiency, as well as the cellular/tissue distribution need to be deeply discussed.

> Thanks for your advice. The relative discussed about the cytotoxicity were presented at the pages10-11, 423-431. And we also give guidance for selecting natural original or low-toxicity polymers such as chitosan, chitosan, arginine, and low molecule weight PEI. The other discussions about transfection efficiency, cellular uptake efficiency in the section 3 and section 4. We firmly discussed the how the types and other characters of materials affected transfer efficiency and cellular uptake efficiency.

  1. In animal cell-based gene transfection, lipofectamine and Polyethyleneimine (PEI) were usually selected as the “gold standards”, is there any materials set as “gold standards” in plant cell gene transfection system?

> Thanks for this question. Different from the animal cell-based experiments, which are relatively simple and have been investigated by many labs, the plant-based transfection experiments are much more difficult since the materials needed to transform is often the explants and we need to obtain whole plants from the transfected cells. What’s more, the application of nanomaterials in plants is still in the initial stage, and there is still no complete and comprehensive comparation of all the materials and experiment conditions in plants, thus there is no golden standards in plant transfection. We have added such discussions in line 399-407. This is also one of the reasons for this review, we wish to give a nice review to push forward the studies, so that we can also have golden standard, and everyone can easily use these techniques. 

  1. The conclusion part is a bit simple; the advantages and disadvantages of current plant gene delivery system should be clearly presented. Moreover, the future perspective in this field needs to be discussed. 

>Thanks for your constructive suggestions. We furtherly revised section 4 and section 5. We have added the comparation and discussion of nanomaterials with traditional method. We have also addressed the future potential and directions for this field. Please refer to line 432-491.

Reviewer 2 Report

The manuscript The Promising Nano-Vectors for Gene Delivery in Plant Genome

Engineeringcontains interesting data, it is well organized and written.

However, there are some points that must be improved, see below:

1) The paragraph at page 2 lines 3-9 contains information too general about nanotechnology. I suggest to be removed or replaced with more relevant considerations about the use of nanotechnology in gene delivery.

2) Page 5, paragraph 3 “It was first reported in 1992 that a mirco-material, silicon” probably is micromaterial.

3) In table 2 term “Clay sheets” must be replaced with the scientific name “Layered

double hydroxides”

4) Table 2 – since the table contain a summary of the nanomaterials used as vectors, I suggest a single registry for Gold nanoparticles with different references and description of the type of particles

5) The main concern is that authors excluded from their manuscript both liposomes and polymeric nanoparticles, which are very promising nanomaterials as gene delivery systems in plant genetic engineering. I strongly suggest to add some paragraphs with relevant results about these subjects.

6) In the section 4, page 8 Paragraph “Transient expressing efficiency is not…” must be extended, since is the most important part of the section.

Author Response

Reviewer 2:

The manuscript The Promising Nano-Vectors for Gene Delivery in Plant Genome Engineering” contains interesting data, it is well organized and written. However, there are some points that must be improved, see below: 

1) The paragraph at page 2 lines 3-9 contains information too general about nanotechnology. I suggest to be removed or replaced with more relevant considerations about the use of nanotechnology in gene delivery.

> Thanks for your nice suggestion. We have added the introduction of genome editing technologies and the great potential of nanomaterials in the delivering the genome editing regents in the introduction part. Please refer to line 68-82

For the purpose of broad interest, we kept the general introduction of nanotechnology.

2) Page 5, paragraph 3 “It was first reported in 1992 that a mirco-material, silicon” probably is micromaterial.

> Thanks for your nice advice. We have corrected this mistake. Please refer to line 210.

3) In table 2 term “Clay sheets” must be replaced with the scientific name “Layered double hydroxides”

> Thanks for your nice advice. We have replaced this according to your suggestion.

4) Table 2 – since the table contain a summary of the nanomaterials used as vectors, I suggest a single registry for Gold nanoparticles with different references and description of the type of particles

> Thanks for your nice advice.  We have added a separate description of gold MSNs ,the different AuNPs (different size, shapes or magnetic) and liposome.

5) The main concern is that authors excluded from their manuscript both liposomes and polymeric nanoparticles, which are very promising nanomaterials as gene delivery systems in plant genetic engineering. I strongly suggest to add some paragraphs with relevant results about these subjects.

> Thanks for your professional suggestion. We have added the introduction of organic polymers and liposomes in this version, please refer to line 327-366. The chitosan and starch also discussed by us.

6) In the section 4, page 8 Paragraph “Transient expressing efficiency is not…” must be extended, since is the most important part of the section.

> Thanks for your nice advice. We have added more details and discussions of the expression efficiency of nanomaterials. Please refer to line 399-423.

Reviewer 3 Report

This is a review manuscript aiming to provide an overview of nanovectors for plant genome engineering. However, the information provided in this review is far from sufficient to make readers get a clear idea of this field, let alone in-depth understanding of this topic. The content should be greatly enriched. Definitely more detailed description should be added to section 3. Representative published papers or work should be highlighted and substantially described. Data related to these representative papers should be presented in the form of figures to explain the results clearly to readers, especially to those who are new to the field. The acceptance of this paper could be re-considered after the manuscript is carefully revised.

Author Response

Reviewer 3:

This is a review manuscript aiming to provide an overview of nano-vectors for plant genome engineering. However, the information provided in this review is far from sufficient to make readers get a clear idea of this field, let alone in-depth understanding of this topic. The content should be greatly enriched. Definitely more detailed description should be added to section 3. Representative published papers or work should be highlighted and substantially described. Data related to these representative papers should be presented in the form of figures to explain the results clearly to readers, especially to those who are new to the field. The acceptance of this paper could be re-considered after the manuscript is carefully revised.

>We thank the reviewer for providing so many nice advices and giving us the opportunity to improve our manuscript. With several nice reviews on the development of nanomaterials and the application of nanomaterials in plants, which we have cited in the introduction part, we wish to give an up-to-date review of the progresses of nanomaterials in plant genetic engineering, especially plant genome editing.

In this version, we have added more detailed descriptions of the application of different nanomaterials in plant genetic engineering. We have also added more deep discussions of these studies, as well as the future direction of this field. We have added some new figures to help the readers understand the nanomaterials can be played an important role in future agricultural activities and breeding works for plants.

Round 2

Reviewer 1 Report

After revision, most of the reviewer’s comments were addressed by the authors in the manuscript. There are small errors need to be corrected before consideration of acceptance:

Conclusion part, page 12, line 469, “The PEG-meothod is a special way for transgene works, the “meothod” should be “method”.; line 471, “foregin gene” should be “foreign gene”.

Author Response

Thanks for your suggestion. We have re-checked the spelling and  grammar bugs of the manuscript. 

Reviewer 2 Report

The manuscript could be published in the revised form.

Author Response

Thank you for your patience in reviewing the manuscript.  We revised the form of the manuscript. 

Reviewer 3 Report

The authors addressed my questions and I have no further questions.

Author Response

Very thanks for your suggestions . We changed the form of our manuscript according to the IJMS principles.